# Generative Category-Level Shape and Pose Estimation with Semantic Primitives

Guanglin Li[1,2]  Yifeng Li[2]  Zhichao Ye[1]  Qihang Zhang[3]
Tao Kong[2]  Zhaopeng Cui[1]  Guofeng Zhang[1]
[1]State Key Lab of CAD&CG, Zhejiang University  [2]ByteDance AI Lab
[3]Multimedia Laboratory, The Chinese University of Hong Kong

**Abstract:** Empowering autonomous agents with 3D understanding for daily objects is a grand challenge in robotics applications. When exploring in an unknown environment, existing methods for object pose estimation are still not satisfactory due to the diversity of object shapes. In this paper, we propose a novel framework for category-level object shape and pose estimation from a single RGB-D image. To handle the intra-category variation, we adopt a semantic primitive representation that encodes diverse shapes into a unified latent space, which is the key to establish reliable correspondences between observed point clouds and estimated shapes. Then, by using a SIM(3)-invariant shape descriptor, we gracefully decouple the shape and pose of an object, thus supporting latent shape optimization of target objects in arbitrary poses. Extensive experiments show that the proposed method achieves SOTA pose estimation performance and better generalization in the real-world dataset. Code and video are available at https://zju3dv.github.io/gCasp.

**Keywords:** Category-level Pose Estimation, Shape Estimation

## 1 Introduction

Estimating the shape and pose for daily objects is a fundamental function which has various applications, including 3D scene understanding, robot manipulation and autonomous warehousing [1, 2, 3, 4, 5, 6]. Early works for this task are focused on *instance-level* pose estimation [7, 8, 9, 10, 11], which aligns the observed object with the given CAD model. However, such a setting is limited in real-world scenarios since it is hard to obtain the exact model of a casual object in advance. To generalize to these unseen but semantically familiar objects, *category-level* pose estimation is raising more and more research attention [12, 13, 14, 15, 16, 17] since it could potentially handle various instances of the same category in real scenes.

Existing works on category-level pose estimation usually try to predict pixel-wise normalized coordinates for instances within one class [12] or adopt a canonical prior model with shape deformations to estimate object poses [14, 15]. Although great advances have been made, these one-pass prediction methods are still faced with difficulties when large shape differences exist within the same category. To handle the variety of intra-class objects, some works [18, 16] leverage neural implicit representation [19] to fit the shape of the target object by iteratively optimizing the pose and shape in a latent space, and achieve better performance. However, in such methods, the pose and shape estimation are coupled together and rely on each other to get reliable results. Thus, their performance is unsatisfactory in real scene (see Sec. 4.2). So we can see that the huge intra-class shape differences and coupled estimation of shapes and poses are two main challenges for the existing category-level pose estimation methods.

To tackle these challenges, we propose to estimate the object poses and shapes with *semantic primitives* from a generative perspective. The insight behind this is that the objects of a category are often composed of components with the same semantics although their shapes are various, *e.g.*, a cup is usually composed of a semicircular handle and a cylindrical body (see Fig. 1(a)). This property

---

Work done during Guanglin Li's internship at ByteDance.

6th Conference on Robot Learning (CoRL 2022), Auckland, New Zealand.

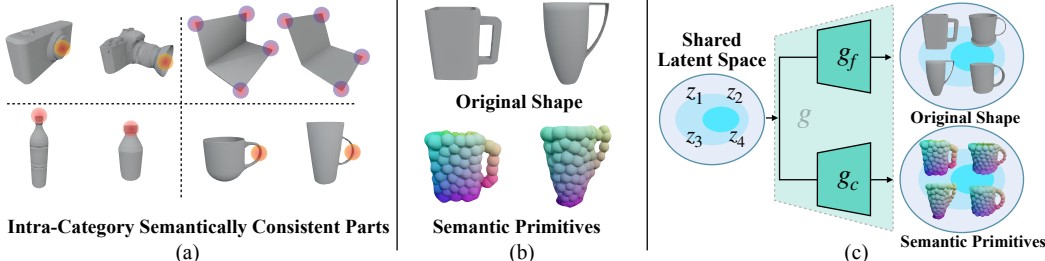

Figure 1: (a) An example of different instances in the same categories, we highlight the same semantic parts (*e.g.*, lens of cameras) between different instances. (b) The detailed shapes (top row) are decomposed into semantic primitives (bottom row). Different colors on primitives indicate different semantic labels, which are consistent among different shapes. (c) An overview of the generative model consisting of two auto-decoders (*i.e.*, $g_f$ and $g_c$) sharing the same latent space. $g_f$ captures fine details, and $g_c$ represents the shapes by simple and semantically consistent shape primitives.

inspired us to handle huge differences in intra-class shapes by fusing this category-level semantic information. Specifically, we propose to encode object shapes into semantic primitives, which construct a unified semantic representation among objects in the same category (see Fig. 1(b)). Following Hao et al. [20], we map diverse shapes and the corresponding semantic primitives into the same shared latent space of a generative model. In this way, we can obtain the correspondence between the observed point cloud and the latent space by dividing the observed point cloud into several semantic primitives with a part segmentation network.

In order to disentangle the shape and pose estimation, we further extract a novel SIM(3)-invariant descriptor from the geometric distribution of the point cloud's semantic primitives. By leveraging such SIM(3)-invariant descriptors, we can perform latent optimization of object shapes in arbitrary poses and obtain the resulting shape consistent with the observation. After obtaining the estimated shape, we use a correspondence-based algorithm to recover the size and pose by aligning semantic primitives of the target object with that of the estimated shape.

In summary, our contributions are as follows: **1)** We propose a novel category-level pose estimation method, which uses semantic primitives to model diverse shapes and bridges the observed point clouds with implicitly generated shapes. **2)** We propose an online shape estimation method, which enables object shape optimization in arbitrary poses with a novel SIM(3)-invariant descriptor. **3)** Even when training on the synthetic dataset only, our method shows good generalization and achieves state-of-the-art category-level pose estimation performance on the real dataset.

## 2 Related Works

**Instance-Level Object Pose Estimation.** Instance-level pose estimation assumes the CAD model of test objects is known in the training phase, and previous methods are mainly divided into three categories. The first category of methods [21, 8, 7, 22] directly regress the 6D object pose from RGB or RGB-D images. PoseCNN [7] extends 2D detection architecture to regress 6D object pose from RGB images, and DenseFusion [8] combines color and depth information from the RGB-D input and makes the regressed pose more accurate. The second are correspondence-based methods [9, 10, 11, 23]. They predict the correspondences between 2D images and 3D models, and then recover the pose by Perspective-n-Point algorithm [24]. For example, PVNet [9] predicts the 3D keypoints on the RGB image by a voting scheme. CDPN [11] predicts the dense correspondences between image pixels and 3D object surface. The third category is rendering-based methods [25, 26], which recover the pose by minimizing the re-projection error between the posed 3D model and 2D image through a differentiable render. Compared with these instance-level methods, our category-level method does not need the known CAD model as prior.

**Category-Level Object Pose Estimation.** Category-level methods [12, 27, 13] aim at estimating the pose for the arbitrary shape of the same category. To overcome the intra-category variations, NOCS [12] introduces normalized object canonical space, which establishes a unified coordinate space among instances of the same category. CASS [27] learns a variational auto-encoder to recover object shape and predicts the pose in an end-to-end neural network. These methods directly regress the pose or coordinates, and thus struggle to form an intra-category representation. Other methods

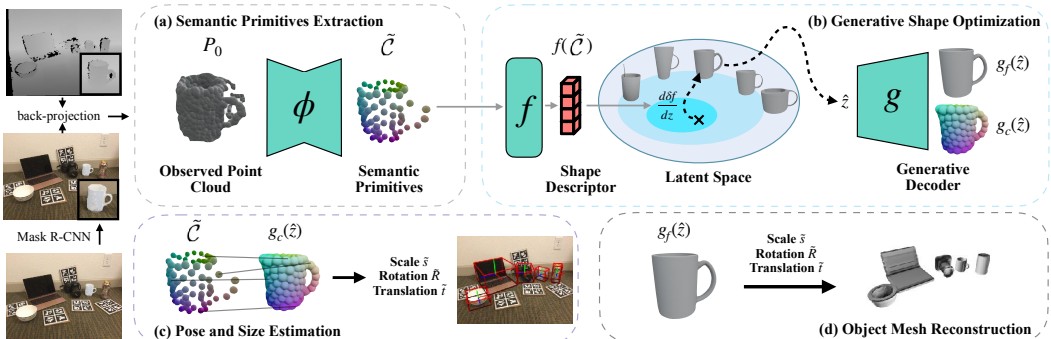

Figure 2: Overview of the proposed method. The input of our method is the point cloud observed from a single-view depth image. (a) Our method first extracts semantic primitives $\tilde{\mathcal{C}}$ from object point cloud $P_0$ by a part segmentation network $\phi$. (b) We calculate a SIM(3)-invariant shape descriptor $f(\tilde{\mathcal{C}})$ from $\tilde{\mathcal{C}}$ and optimize a shape embedding $\hat{z}$ in the latent space by the difference between $f(\tilde{\mathcal{C}})$ and $f(g_c(z))$, where $g_c$ and $g_f$ are the coarse and fine branches of the generative model $g$, detailed in Sec. 3.1. (c) The similarity transformation $\{\tilde{s}, \tilde{\boldsymbol{R}}, \tilde{\boldsymbol{t}}\}$ is recovered through the semantic correspondences between $\tilde{\mathcal{C}}$ and optimized shape $g_c(\hat{z})$. (d) Further, we can simply apply the transformation $\{\tilde{s}, \tilde{\boldsymbol{R}}, \tilde{\boldsymbol{t}}\}$ on the fine geometry generated through $g_f(\hat{z})$, and reconstruct the object-level scene as a by-product besides our main results, which is detailed in the Sec. B in the supp. material.

utilize a prior model of each category for the category-level representation. For example, SPD [14] and SGPA [15] predict deformation on canonical prior models and recover the 6D pose through correspondence between the deformed model and the observed point cloud. However, when facing a significant difference between the prior model and the target object, the deformation prediction tend to fail. DualPose [28] takes the consistency between an implicit pose decoder and an explicit one into account and refines the predicted pose during testing. Besides, FS-Net [13] proposes a novel data augmentation mechanism to overcome intra-category variations. SAR-Net [17] recovers the shape by modeling symmetric correspondence of objects, which makes pose and size prediction easier. To utilize the diverse and continuous shapes in generative models, iCaps[16] and ELLIPSDF[18] introduce the generative models [19, 20] to their shape and pose estimation. Shapes and poses are jointly optimized in these works. Unlike these previous works, we establish a semantically consistent representation and decouple the shape and pose estimation, which makes the optimization more robust.

## 3 Method

**Problem Formulation.** Given observed point cloud $P_0 = \{\boldsymbol{p}_i | i = 1, ..., N_0\}$ of an object with known category, our goal is to estimate the complete shape, scale and 6D pose for this object. The recovered shape is represented by a shape embedding $\hat{z}$ in a latent space. The estimated scale and 6D pose are represented as a similarity transformation $\{s, \boldsymbol{R}, \boldsymbol{t}\} \in \text{SIM}(3)$, where scale $s \in \mathbb{R}$, 3D rotation $\boldsymbol{R} \in \text{SO}(3)$ and 3D translation $\boldsymbol{t} \in \mathbb{R}^3$. SIM(3) and SO(3) indicate the Lie group of 3D similarity transformations and 3D rotation, respectively.

**Overview.** As illustrated in Fig 2, our method takes the RGB-D image as input. A Mask-RCNN[29] network is used to obtain the mask and category of each object instance. In the following stages, our method only processes the observed instance point cloud back-projected from the masked depth image. To overcome the huge intra-category variations, we utilize a unified intra-category representation of semantic primitives (Sec 3.1). Then, we process the observed point cloud with a part segmentation network to extract semantic primitives which establish a connection between the observed object and the latent space (Sec 3.2). To estimate the shape of the target with its pose unknown, we design a SIM(3)-invariant descriptor to optimize the generative shape in the latent space (Sec 3.3). Finally, we estimate the 6D pose and scale of the object through the semantic correspondence between the observed point cloud and our recovered generative shape (Sec 3.4).

## 3.1 Semantic Primitive Representation

The huge intra-category differences make it challenging to represent objects with a standard model. To overcome this problem, we utilize a semantic primitive representation. The insight behind the semantic primitive representation is that although different instances in the same category have various shapes, they tend to have similar semantic parts, *e.g.*, each camera instance has a lens, and each laptop has four corners (see Fig 1(a)). In semantic primitive representation, each instance is decomposed into several parts, and each part always corresponds to the same semantic part in different instances (see Fig 1(b)). To generate this representation, we learn a generative model $g$ following the network structure in [20] and map instances of the same category into a latent space.

Specifically, Fig 1(c) gives an overview of the generative model. The generative model $g$ expresses shapes at two granular levels, one capturing detailed shapes and the other representing an abstracted shape using semantically consistent shape primitives. We note the fine-scale model as $g_f$ and the coarse-scale one as $g_c$. $g_f$ and $g_c$ are two auto-decoders sharing the same latent code $z$. $g_f(z, \boldsymbol{x}) = SDF(\boldsymbol{x})$, where $\boldsymbol{x}$ is a 3D position in canonical space and $SDF$ denotes the signed distance function. $g_c(z) = \{\boldsymbol{\alpha}_i | i = 1, ..., N_c\}$, where $N_c$ is the number of primitives and $\boldsymbol{\alpha}_i$ is the parameters of a primitive. We use sphere primitives here and $\boldsymbol{\alpha} = (\boldsymbol{c}, r)$, where $\boldsymbol{c}$ is the 3D center of a sphere and $r$ is the radius. Please refer to [20] and Sec. A.3 in the supp. material for more details.

## 3.2 Extract Semantic Primitives from Point Cloud

We use a part segmentation network to obtain the semantic primitives of the observation point cloud in two stages. First, we predict a semantic label $l_i$ for each point $\boldsymbol{p}_i$ in the observed point cloud $P_0$, and then a centralization operation is performed on the points to extract the center of each primitive.

**Semantic Label Prediction.** For each point $\boldsymbol{p}_i \in P_0$, its semantic label $l_i$ is defined in terms of closest primitive center $\boldsymbol{c}_j$:

$$l_i = \operatorname*{argmin}_{j=1...N_c} ||\boldsymbol{p}_i - \boldsymbol{c}_j||_2. \tag{1}$$

We treat point label prediction as a classification problem and follow the part segmentation version of 3D-GCN [30] to perform point-wise classification. Standard cross entropy is used as the loss function. In addition, because of imperfect segmentation masks produced by Mask R-CNN, the point cloud back-projected from the masked depth image may contain extra background points. For these outlier points, we add another "dust" label in the point-wise classification to filter out them.

**Symmetric Loss.** Many common objects in robot manipulation have symmetrical structures, and most objects (*e.g., bottles*) are symmetrical about an axis. The unique pre-labeled ground truth cause ambiguity when the object rotates around the axis. In order to eliminate the influence of symmetric structure on our classification task, we introduce the symmetric loss as the same as [12]. For each symmetric object in the dataset, we define a symmetric axis and make the loss identical when the object rotates around the axis. Specifically, we generate a separate ground truth for each possible angle of rotational symmetry. Given the rotation angle $\theta$, the rotated semantic label $l_{\theta,i}$ is defined as:

$$l_{\theta,i} = \operatorname*{argmin}_{j=1...N_c} ||\boldsymbol{p}_i - \boldsymbol{c}_{\theta,j}||_2, \tag{2}$$

where $\boldsymbol{c}_{\theta,j}$ is the position of $\boldsymbol{c}_j$ after $\theta$ degrees of rotation about the axis. Then we define the symmetric loss $L_{sym}$ as:

$$L_{sym} = \min_{\theta \in \Theta} L(\tilde{l}_i, l_{\theta,i}). \tag{3}$$

where $L$ is the standard cross entropy, $\tilde{l}_i$ is the predicted label by the network, and we set $\Theta = \{0°\}$ for non-symmetric objects and $\Theta = \{i \cdot 60° | i = 0, ..., 5\}$ for symmetric objects in our experiments.

**Primitive centralization.** After predicting the semantic label of each 3D point, we count the semantic labels $\tilde{\mathcal{L}} = \{l_i | i = 1, ..., \tilde{N}_c\}$ which appear in the partially observed point cloud. Then, we calculate the primitive centers $\tilde{\mathcal{C}} = \{\boldsymbol{c}_l | l = l_1, ..., l_{\tilde{N}_c}\}$ from the labeled points by averaging the points with the same label. Specifically,

$$\boldsymbol{c}_l = \frac{1}{N_l} \sum_{i=1}^{N_0} \boldsymbol{p}_i \mathbb{I}(\tilde{l}_i = l). \tag{4}$$

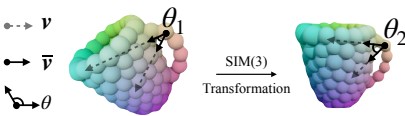

Figure 3: The proposed atomic SIM(3)-invariant descriptor. The angle $\theta_1$ and $\theta_2$, which we called atomic descriptors are invariant with different object poses. And our shape descriptor concatenated by multiple atomic descriptors is also invariant.

where $N_l$ is the total number of points with predicted label $l$, $\mathbb{I}$ is an indication function indicating whether the predicted label $\tilde{l}_i$ equals $l$.

### 3.3 Shape Optimization with SIM(3)-Invariant Descriptor

We decouple the shape estimation from the pose estimation to perform shape estimation for the observed object when the pose is unknown. In order to achieve this pose-invariant shape optimization, we propose a novel SIM(3)-invariant descriptor extracted from semantic primitive representation.

**SIM(3)-invariant Descriptor.** Given a set of 3D points $\mathcal{C} = \{c_i | i = 1, ..., N\}$ and a SIM(3)-transformation $\{s, \boldsymbol{R}, \boldsymbol{t}\}$, a SIM(3)-invariant descriptor $f_0$ can be expressed as:

$$f_0(\mathcal{C}) = f_0(s\boldsymbol{R}\mathcal{C} + \boldsymbol{t}). \tag{5}$$

To make our descriptor SIM(3)-invariant, first, we obtain a translation-invariant descriptor by calculating the vectors $\boldsymbol{v}$ composed of every two points:

$$\boldsymbol{v}_{ij} = (\boldsymbol{c}_i + \boldsymbol{t}) - (\boldsymbol{c}_j + \boldsymbol{t}) = \boldsymbol{c}_i - \boldsymbol{c}_j. \tag{6}$$

After that, we normalize the vector $\boldsymbol{v}$ to eliminate the scale factor $s$:

$$\overline{\boldsymbol{v}}_{ij} = \frac{s\boldsymbol{v}_{ij}}{||s\boldsymbol{v}_{ij}||_2} = \frac{\boldsymbol{v}_{ij}}{||\boldsymbol{v}_{ij}||_2}. \tag{7}$$

Finally, we eliminate the effect of rotation transformation by calculating the dot product between two vectors (see Fig. 3):

$$\theta(\boldsymbol{c}_i, \boldsymbol{c}_j, \boldsymbol{c}_k, \boldsymbol{c}_o) = \boldsymbol{R}\overline{\boldsymbol{v}}_{ij} \cdot \boldsymbol{R}\overline{\boldsymbol{v}}_{ko} = \overline{\boldsymbol{v}}_{ij}^T \boldsymbol{R}^T \boldsymbol{R}\overline{\boldsymbol{v}}_{ko} = \overline{\boldsymbol{v}}_{ij}^T \overline{\boldsymbol{v}}_{ko}, \tag{8}$$

where $\cdot$ denotes the dot product of two vectors, and $\theta$ denotes a function which takes 4 points as input and a SIM(3)-invariant scalar as output (as the angles shown in Fig. 3)

We call $\theta$ the atomic descriptor. However, only a single atomic descriptor calculated from 4 points (primitives centers) is not enough to describe an object's shape. Thus, we design the SIM(3)-invariant shape descriptor through a function $f$, which concatenates multiple atomic descriptors to form a SIM(3)-invariant shape descriptor. With our estimated primitives $\tilde{\mathcal{C}}$ and their semantic labels $\tilde{\mathcal{L}}$, the set $\tilde{\mathcal{L}}^4$ is denoted as:

$$\tilde{\mathcal{L}}^4 = \{(i, j, k, o) \mid i, j, k, o \in \tilde{\mathcal{L}}, i \neq j, k \neq o\}, \tag{9}$$

and the SIM(3)-invariant shape descriptor $f$ is denoted as:

$$f(\tilde{\mathcal{C}}, \tilde{\mathcal{L}}) = \bigoplus_{(i,j,k,o) \in \tilde{\mathcal{L}}_{sub}^4} \theta(\boldsymbol{c}_i, \boldsymbol{c}_j, \boldsymbol{c}_k, \boldsymbol{c}_o), \tag{10}$$

where $\oplus$ denotes the concatenate operation which concatenates multiple scalars into a vector in order, and $\tilde{\mathcal{L}}_{sub}^4$ is a subset randomly chosen from $\tilde{\mathcal{L}}^4$ for each optimization. We choose the subset of $\tilde{\mathcal{L}}^4$ because $|\tilde{\mathcal{L}}^4|$ would be large, which is computational expensive. In Sec. 4.3, an ablation is conducted to analyze the effect of $|\tilde{\mathcal{L}}_{sub}^4|$, which indicates the number of atomic descriptors in a shape descriptor.

**Online Shape Optimization.** Given the shape embedding $\hat{z}$ in the latent space and estimated primitives $\tilde{\mathcal{C}}$ with semantic labels $\tilde{\mathcal{L}}$, we can measure the shape difference between the generative shape $g_c(\hat{z})$ and the observation $P_0$ through the SIM(3)-invariant descriptor. We denote this difference as $\tilde{e}$:

$$\tilde{e} = ||f(\tilde{\mathcal{C}}, \tilde{\mathcal{L}}) - f(\hat{\mathcal{C}}(g_c(\hat{z})), \tilde{\mathcal{L}})||_2 + \eta||\hat{z}||_2, \tag{11}$$

where $\hat{\mathcal{C}}(g_c(\hat{z})) = \{\hat{c}_i | i = 1, ..., N_c\}$ is the primitive centers of $g_c(\hat{z})$. Obviously, $f$ is a differentiable function and $g_c$ is a differentiable neural network. We optimize the shape embedding $\hat{z}$ via gradient descent in an online iterative way and set $\eta = 0.0001$ to weight the regularization term.

Table 1: Comparison of our method with other five SOTA methods [12, 14, 15, 17, 16] on REAL275 and CAMERA25. 'S' is synthetic data and 'R' is real data.

| Dataset | Methods | Training Data | Canonical Model | IoU50 | IoU75 | $5°2cm$ | $5°5cm$ | $10°2cm$ | $10°5cm$ | Learnable Parameters(M) |
|---|---|---|---|---|---|---|---|---|---|---|
| REAL275 (real) | NOCS [12] | RGB(S+R) | Regress | 78.0 | 30.1 | 7.2 | 10.0 | 13.8 | 25.2 | - |
| | SPD [14] | RGBD(S+R) | Deform | 77.3 | 53.2 | 19.3 | 21.4 | 43.2 | 54.1 | 18.3 |
| | SGPA [15] | RGBD(S+R) | Deform | **80.1** | 61.9 | 35.9 | 39.6 | 61.3 | 70.7 | 23.3 |
| | SAR-Net [17] | D(S) | Deform | 79.3 | 62.4 | 31.6 | 42.3 | 50.3 | 68.3 | 6.3 |
| | iCaps [16] | D(S) | Generate | - | - | - | 22.3 | - | - | 80.9 |
| | Ours | D(S) | Generate | 79.0 | **65.3** | **46.9** | **54.7** | **64.2** | **76.3** | **2.3** |
| CAMERA25 (synthetic) | NOCS [12] | RGB(S+R) | Regress | 83.9 | 69.5 | 32.3 | 40.9 | 48.2 | 64.6 | - |
| | SGPA [15] | RGBD(S+R) | Deform | 93.2 | 88.1 | 70.7 | 74.5 | **82.7** | **88.4** | 23.3 |
| | SAR-Net [17] | D(S) | Deform | 86.8 | 79.0 | 66.7 | 70.9 | 75.3 | 80.3 | 6.3 |
| | Ours | D(S) | Generate | **95.7** | **89.3** | **71.7** | **77.0** | 80.1 | 86.9 | **2.3** |

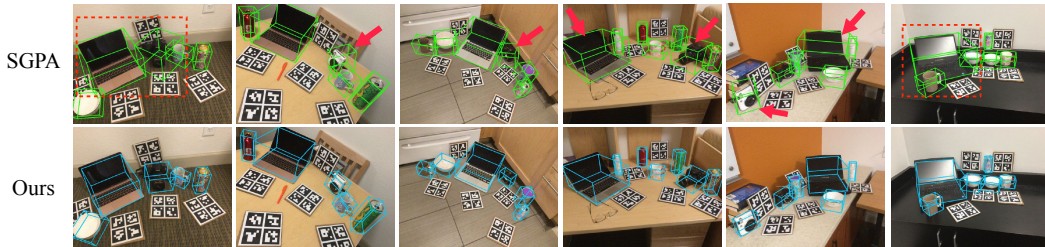

Figure 4: Qualitative comparisons between our method and SGPA [15] on REAL275 dataset [12]. The red arrows and dotted box point out the inaccurate results of SGPA compared with our method.

### 3.4  6D Pose and Size Estimation

With the primitive centers $\hat{\mathcal{C}}(g_c(\hat{z}))$ of the optimized shape and the observed point cloud, we use Umeyama algorithm [31] to compute the SIM(3) transformation by minimizing following error function:

$$\tilde{s}, \tilde{\boldsymbol{R}}, \tilde{\boldsymbol{t}} = \underset{s, \boldsymbol{R}, \boldsymbol{t}}{\operatorname{argmin}} \frac{1}{N_0} \sum_{i=1}^{N_0} ||\boldsymbol{p}_i - s\boldsymbol{R}(\hat{\boldsymbol{c}}_{l_i} + \boldsymbol{t})||^2. \tag{12}$$

## 4  Experiments

### 4.1  Datasets and Evaluation Metrics

We evaluate our method on NOCS [12] dataset which consists of six objects categories including *bottle, bowl, camera, can, laptop* and *mug*. The NOCS dataset contains a virtual dataset called CAMERA25 and a real-world dataset called REAL275. The CAMERA25 dataset contains 300K synthetic images and there are 25K images for evaluation among them. The REAL275 contains 4.3K real-scene images for training and 2.75K real-scene images for evaluation.

Two aspects of metrics are adopted for evaluation: 3D IoU and pose error. 3D IoU measures the overlap of two posed 3D bounding boxes, and we compute the average precision at threshold values of 50% and 75%. The pose error measures the rotation and translation errors between the predicted pose and ground truth pose, and we compute the average precision at threshold values $(5°, 2cm)$, $(5°, 5cm)$, $(10°, 2cm)$ and $(10°, 5cm)$.

### 4.2  Comparison with State-of-the-Art Methods

We compared our proposed method with several SOTA methods [12, 14, 15, 17, 16]. NOCS [12] directly regress pixel-level canonical coordinate. SPD [14] and SGPA [15] are the methods which recover canonical shape by deforming a prior shape. iCaps [16] and our methods utilize a generative model to refine and recover the canonical model respectively. Table 1 shows the quantitative results. Overall, our method needs the minimum parameters learned for the shape and pose estimation task and shows the results of the smallest domain gap between the synthetic dataset and

Table 2: Pose estimation results with recovered shape by mean, random and optimized shape embedding on REAL275.

| Embedding | IoU50 | IoU75 | 5°2cm | 5°5cm | 10°2cm | 10°5cm |
|---|---|---|---|---|---|---|
| Mean | 77.7 | 54.5 | 31.7 | 36.0 | 51.4 | 61.0 |
| Random | 76.8 | 51.6 | 28.7 | 34.4 | 47.6 | 60.3 |
| Optimized | **79.0** | **65.3** | **46.9** | **54.7** | **64.2** | **76.3** |

Table 3: Results of our method with different numbers of primitives on REAL275

| Number | IoU50 | IoU75 | 5°2cm | 5°5cm | 10°2cm | 10°5cm |
|---|---|---|---|---|---|---|
| 64 | **79.2** | 63.7 | 41.2 | 51.4 | 58.9 | 74.9 |
| 128 | 78.9 | 64.5 | 43.5 | 51.2 | 61.2 | 74.6 |
| 256 | 79.0 | **65.3** | **46.9** | **54.7** | **64.2** | **76.3** |
| 512 | 77.2 | 63.3 | 45.7 | 53.6 | 61.7 | 74.4 |

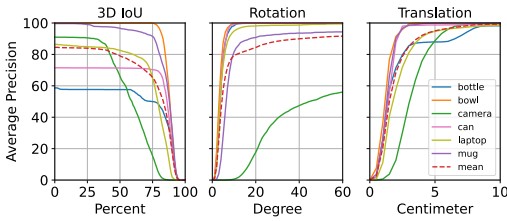
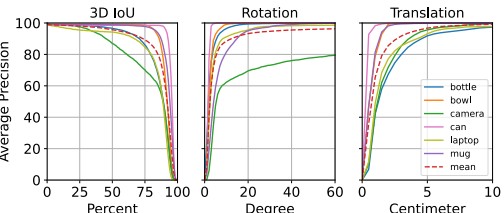

Figure 5: Curves of each category under different thresholds on IoU, rotation and translation on REAL275 (left) and CAMERA (right) dataset.

the real-world dataset. In detail, our method outperforms other existing methods on all metrics but IoU50 on the real-world REAL275 dataset. Fig. 4 shows qualitative comparisons with SGPA [15]. On the synthetic CAMERA25 dataset, our method also shows improvement on most metrics compared with other methods, except for a small gap with SGPA [15] on two metrics, $(10°, 2cm)$ and $(10°, 5cm)$. Besides, Fig. 5 shows the average precision curves of each category under different thresholds on IoU, rotation and translation on REAL275 and CAMERA25 dataset. Compared to the other generative method iCaps [16], our method shows far better results, indicating the advantage of our decoupled shape and pose estimation. It should be noted here that our trained generative model (Sec. 3.1) consists of about 4.3M parameters for each category. The generative models are only pre-trained on ShapeNetCore [32] and need no more fine-tuning for the specific shape and pose estimation task. Thus, we compare the *learnable parameters* with other methods. Specifically, our method needs only additionally train a part segmentation network (2.3M) for all categories on the synthetic training data provided by CAMERA25 [12], which leads to our better generalization in real scenes.

## 4.3 Ablation Study

To justify the design choices of our method, we explore the following issues through experiments:

- Effectiveness of the SIM(3)-invariant descriptor. Does the descriptor-based shape optimization make the optimized shape more accurate and benefit the pose estimation?
- Effect of different numbers of primitives. Semantic primitives are the basis of our method. How many primitives are better for pose estimation?
- Effect of different numbers of atomic descriptors $|\tilde{\mathcal{L}}^4_{sub}|$. Our shape descriptor is concatenated by multiple atomic descriptors. How many atomic descriptors are more effective for shape optimization?

**Effectiveness of SIM(3)-invariant descriptors.** We verify the effectiveness of our SIM(3)-invariant descriptors in two aspects. First, we verify that descriptor-based shape optimization improves the accuracy of shape reconstruction. We compare the reconstruction error of our optimized shape with the shape when the shape embedding $z = \mathbf{0}$ (denoted as "Mean" shape) and a random shape when the shape embedding $z \sim \mathcal{N}(\mathbf{0}, \mathbf{1})$ (denoted as "Random" shape). Table 4 shows that our optimized shapes are more accurate. In Table 4, we also compare our method with SGPA [15], even without shape loss to supervise our method, our coarse-level shape reconstruction shows comparable results with SGPA. Second, we verify that descriptor-based shape optimization improves the pose estimation results. We compare the results of the pose estimation under the three canonical shapes (mean, random and optimized) in the Table 2. The results show that we get far better pose estimation results based on the optimized shape. Besides, Fig. 6 visualizes cases where shapes become more consistent with observations as optimization progresses.

Table 4: Comparison of shape reconstruction accuracy of our methods and SGPA in CD metric ($\times 10^{-3}$) on REAL275. Mean and Random refer to the shape when shape embedding $z = \mathbf{0}$ and $z \sim \mathcal{N}(\mathbf{0}, \mathbf{1})$.

| Methods | bottle | bowl | camera | can | laptop | mug | all |
|---------|--------|------|--------|-----|--------|-----|-----|
| SGPA [15] | 2.93 | **0.89** | **5.51** | 1.75 | **1.62** | **1.12** | **2.44** |
| Mean | 9.55 | 4.39 | 12.4 | 3.03 | 6.59 | 2.26 | 6.05 |
| Random | 11.6 | 5.49 | 16.1 | 3.74 | 7.12 | 3.63 | 7.56 |
| Optimized | **2.05** | 1.55 | 10.1 | **1.63** | 2.12 | 2.93 | 3.46 |

Table 5: Comparison of shape reconstruction accuracy when different number of atomic primitives are randomly chosen to optimize the shape in CD ($\times 10^{-3}$) metric.

| $|\tilde{\mathcal{L}}_{sub}^4|$ | w/o. RANSAC | w. RANSAC |
|---------|-------------|-----------|
| $10^1$ | $6.79 \pm 6.50$ | $4.38 \pm 5.41$ |
| $10^2$ | $4.86 \pm 5.38$ | $3.97 \pm 5.15$ |
| $10^4$ | $3.68 \pm 4.34$ | $3.59 \pm 4.54$ |
| $10^6$ | $3.46 \pm 4.05$ | $3.44 \pm 4.13$ |

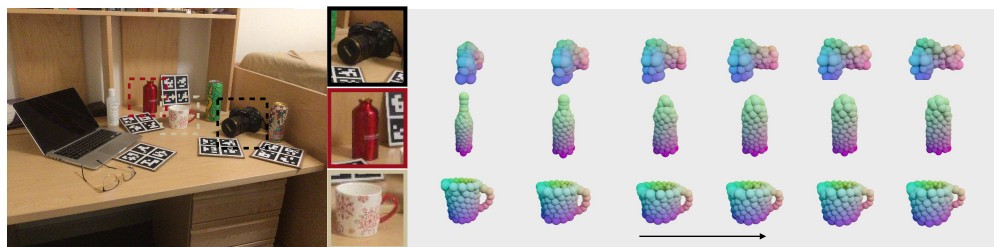

Real Scene      Target Objects      Iterative Optimization

Figure 6: Visualization of the shape optimization process. As optimization progresses, our optimized shapes are more consist with observations.

**Effect of different numbers of primitives.** We conduct an ablation study on the number of semantic primitives. We experiment the number with 64, 128, 256, and 512. As shown in Table 3, the results show that the accuracy of pose estimation is improved with the increase of the number of primitives, which indicates that more primitives can represent more complex geometry. The results show a decline when the number is set to 512 because 512 primitives are relatively redundant for our 1024 input points, and more primitive classes also decline the accuracy of part segmentation network (Sec. 3.2). Based on these observations, we use 256 primitives as the default setting in other experiments.

**Effect of different numbers of atomic descriptors.** As illustrated in Sec. 3.3, a single atomic descriptor is insufficient to describe an object's shape, and concatenating all atomic descriptors to form a shape descriptor would be computationally expensive and unnecessary. Thus, we conduct experiments and analyze the influence of the number of atomic descriptors on the shape optimization results. Specifically, we randomly choose $\{10^1, 10^2, 10^4, 10^6\}$ atomic descriptors to form the shape descriptor and use the chamfer distance to evaluate the quality of the optimized shape. As shown in Table. 5, the reconstruction error of the optimized shape decreases as the number of atomic descriptors increases, and applying RANSAC in the random selection can further reduce this error. In the main experiments, we choose $10^6$ atomic descriptors depending on the computational resources and do not use RANSAC for efficiency.

## 5    Conclusion and Limitations

We propose a novel category-level shape and pose estimation framework by utilizing the semantic primitive representation. Leveraging a novel SIM(3)-invariant descriptor, we decouple shape and pose estimation and optimize the implicitly generated shape in an online iterative way. Experiments results demonstrate the advantages of our method on the real-scene data compared with other SOTA methods.

**Limitations.** As other methods [14, 15, 17], our method also relies on the pre-processed masks of object instances by Mask R-CNN and extra annotation of symmetry axes when training. Furthermore, although we can reconstruct the object mesh (Sec. B in the supp. material), this by-product has no influence or contribution to our work. Future work could consider the mesh reconstruction to get better shape and pose estimation results. Besides, our method does not consider the occlusion and collision between objects, and may cause ambiguity from single-view observation. Feasible future work could fuse color information into our input, or expand the proposed method to multi-view observation to address these problems.

**Acknowledgments**

We would like to thank Bangbang Yang for the useful discussion on this paper.

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
