# OpenReview forum: "Generative Category-Level Shape and Pose Estimation with Semantic Primitives"
_robot-learning.org/CoRL/2022/Conference — CoRL 2022 Poster_

### Official Review · Reviewer_imVj · 2022-07-05

**Originality:** Very Good
**Technical Quality:** Very Good
**Clarity Of Presentation:** Excellent
**Impact:** 3

**Recommendation:**

Weak Accept: I recommend accepting the paper, but will not argue for my recommendation if the majority of other reviewers have a different opinion.

**Summary:**

This paper proposes a method for joint shape and pose estimation for categories of objects. The method disentangles the two estimation tasks by a proposed SIM(3)-invariant shape descriptor, combined with using DualSDF (Hao et al., 2020), for shape estimation. On the NOCS dataset, the paper demonstrates the usefulness of the disentangled estimation approach, showing improved performance compared to state-of-the-art category-level pose estimation methods.

**Issues:**

Please address the points mentioned in "weaknesses" and "questions" above, in that order of priority.

**Quality Of The Limitations Section:**

Limitations are addressed clearly

**Reviewer Expertise:**

4: The reviewer is confident but not absolutely certain that the evaluation is correct

**Robotics Focus:**

Highly relevant to robotics but no hardware experiments

**Strengths And Weaknesses:**

## Strengths
The main innovation of the paper is the SIM(3)-invariant shape descriptor used in conjunction with combined shape estimation and pose estimation. The methods are clearly described. The writing in the paper is clear, and it is overall a pleasurable read. The results on the REAL275 subset of the NOCS dataset are encouraging, especially as much fewer parameters are needed than competing methods to reach competitive results.


## Weaknesses

- The evaluation scores chosen ignore object symmetry. Reported scores may look arbitrarily better or worse depending on which symmetry mode the methods predict. The paper should provide data for other metrics such as ADD-S (see, e.g., PoseCNN) that handle symmetry. This would also be a more realistic evaluation for applications where rotational symmetry can be ignored. *Updated after author response*: the authors have clarified the evaluation does consider symmetry.

- The pose and shape estimation aspects are coupled in the evaluation. It is not clear, how much the non-perfect models influence pose estimation performance. Also open is the question of how well the proposed decoupling of pose and shape estimation really works, or whether the networks simply co-adapted to work around each others' deficiencies. Perhaps an ablation by evaluating on instance-level 6D pose estimation, where the ground truth models are available, could be helpful. *Updated after author response*: the new ablation study addresses this concern.

- The training process in the proposed method relies on the ShapeNet CAD models for training the generative models. How could these findings be translated to the case where no CAD models are available? In which domains could one expect this to be critical?  To be clear, I think this limitation is shared by many of the category-level pose estimation works, and not specific to this work. However, some discussion would be welcome as this is also not mentioned in the limitations. *Updated after author response*: in their response, the authors provided comments on this.


## Questions
- How does the representation in Section 3.1 differ from DualSDF? If DualSDF representation is directly adopted, as is the case as far as I can tell, it would be better to remove language such as "we propose" that imply novelty. Otherwise, a clear statement indicating the difference to Hao et al. is needed.

- More details and clarification on the SIM(3)-invariant descriptor and in particular Eq. (9) are needed.
* Is the "concatenate" operator taken over all unique 4-tuples of semantic labels?
* It seems that the descriptor \theta is invariant to the labels \tilde{\mathcal{L}} - why is this the case? Why do we need the label set as an argument of f?

- Sect 3.5: "We sample 1024 points from..." - do you sample the points uniformly at random, or by another method? I suggest to try in the future farthest point sampling to guarantee uniform coverage of the input.


## Minor
- When referencing the supplementary material in the main paper, it would be helpful to always indicate a specific section in the supplement.
- \tilde{l}_i in Eq. (3) is not defined. Is this the label predicted by the network?

**Summary Of Recommendation:**

This paper combines implicit shape representations with a novel SIM(3)-invariant descriptor in an interesting way for category-level pose estimation, and shows some promising results. In their response, the authors provided an ablation study and addressed my other comments. I recommend accepting the paper.

---

> ### Author Response · Authors · 2022-08-23
> **Response to Reviewer jxdo (1/2)**
>
> Thanks for your review and comments. We hope our response addresses your concerns. The response contains two parts. This is the first part.
> 1. The evaluation scores chosen ignore object symmetry.
> - Sorry for not mentioning this information in the paper, but in fact, the symmetry of the objects has been taken into account in the evaluation of all metrics. We have used the evaluation code provided by NOCS to perform a fair comparison.
> 2. The pose and shape estimation aspects are coupled in the evaluation. It is not clear, how much the non-perfect models influence pose estimation performance. Also open is the question of how well the proposed decoupling of pose and shape estimation really works, or whether the networks simply co-adapted to work around each others' deficiencies. Perhaps an ablation by evaluating on instance-level 6D pose estimation, where the ground truth models are available, could be helpful.
> - In the ablation, we compare the pose estimation results of our optimized, mean and random shapes. Mean and random refer to the non-perfect shapes when shape embedding $z=0$ and $z\sim \mathcal{N}(0,1)$ respectively. Compared to these non-perfect models, our optimized shapes significantly improve the results of pose estimation. You could check ["Response to Common Concerns"](https://openreview.net/forum?id=N78I92JIqOJ&noteId=j4LblaHIZDx) for detail.
> - As for decoupling the pose and shape estimation , we think it is difficult to illustrate the benefits through experiments, since our framework is novel and hard to compare with other methods component by component. However, as far as we concerned, it makes sense to decouple the shape estimation from the pose optimization, which is non-linear, so that the number of variables optimized simultaneously is reduced.
> 3. The training process in the proposed method relies on the ShapeNet CAD models for training the generative models. How could these findings be translated to the case where no CAD models are available?
> - From our point of view, the CAD models provide intra-category shape prior and canonical pose for the category-level pose estimation task. The canonical pose is crucial because the estimated pose describes the transformation between the observation and the canonical pose. Pose estimation cannot be defined without a canonical pose. Therefore, without CAD models, we think the most important thing is how to define the canonical pose. We have some ideas.
> - First, if there is enough training data with annotations, the reference model can still be reconstructed from the training data. Many one-shot pose estimation works, such as LatentFusion [a] and OnePose [b], reconstruct the canonical shape from some annotated reference images.
> - Second, we can define canonical pose with intra-category semantic information instead of CAD models. Take the laptop as an example. If we define the canonical pose as the pose when the four corners of the laptop keyboard are in the position of 3D coordinate (0,0,0), (1,0,0), (1,1,0), (0,1,0). Then, we just need a corner detector to extract the four corners of the observed laptop keyboard to estimate the laptop's pose. This idea is similar to our mentioned intra-category semantic consistency.
>
> References:
>
> [a] Park K, Mousavian A, Xiang Y, et al. Latentfusion: End-to-end differentiable reconstruction and rendering for unseen object pose estimation[C]//Proceedings of the IEEE/CVF conference on computer vision and pattern recognition. 2020: 10710-10719.
>
> [b] Sun J, Wang Z, Zhang S, et al. OnePose: One-Shot Object Pose Estimation without CAD Models[C]//Proceedings of the IEEE/CVF Conference on Computer Vision and Pattern Recognition. 2022: 6825-6834.

---

> > ### Comment · Reviewer_imVj · 2022-08-25
> > **Follow-up question & thank you**
> >
> > Thank you for the response. My main concern on pose and shape estimation coupling and the related ablation is resolved by the reply. The comments on the hypothetical case without a CAD model are also reasonable, and perhaps worth considering in future work. I also appreciate the clarifications on my other questions.
> >
> > I do have one question remaining. I didn't quite understand from your response number 1 how the evaluation takes symmetry into account. Could you either explain this or point me to a source where this is explained?

---

> > > ### Author Response · Authors · 2022-08-25
> > > **Response to the follow-up question**
> > >
> > > We are glad that we have addressed your concerns.
> > >
> > > NOCS [a] has a rough description in the paper:
> > > > For symmetric object categories (bottle, bowl, and can), we allow the predicted 3D bounding box to freely rotate around the object's vertical axis with no penalty. We perform special processing for the mug category by making it symmetric when the handle is not visible since it is hard to judge its pose in such cases, even for humans. We use [50] to detect handle visibility for CAMERA data and manually annotate for real data.
> > >
> > > You can also check the implementation details in the open source code, it is quite clear:
> > > - For IoU: https://github.com/hughw19/NOCS_CVPR2019/blob/master/utils.py#L274
> > > - For Pose: https://github.com/hughw19/NOCS_CVPR2019/blob/master/utils.py#L389
> > >
> > > References:
> > >
> > > [a] Wang, He, et al. "Normalized object coordinate space for category-level 6d object pose and size estimation." Proceedings of the IEEE/CVF Conference on Computer Vision and Pattern Recognition. 2019.
> > >
> > > [50] Yi, Li, et al. "A scalable active framework for region annotation in 3d shape collections." ACM Transactions on Graphics (ToG) 35.6 (2016): 1-12.

---

> > > > ### Comment · Reviewer_imVj · 2022-08-25
> > > > **Thanks**
> > > >
> > > > Thank you for the clarification!

---

> ### Author Response · Authors · 2022-08-23
> **Response to Reviewer imVj （2/2）**
>
> This is the second part.
>
> 4. How does the representation in Section 3.1 differ from DualSDF? If DualSDF representation is directly adopted, as is the case as far as I can tell, it would be better to remove language such as "we propose" that imply novelty. Otherwise, a clear statement indicating the difference to Hao et al. is needed.
> - Thanks for your kind reminder. Although we have made a small improvement based on DualSDF (A.3 in the appendix), the core idea has not changed compared with DualSDF. We appreciate your suggestion and change "propose" to "adopt"
> 5. More details and clarification on the SIM(3)-invariant descriptor and in particular Eq. (9) are needed.
> 6. Is the "concatenate" operator taken over all unique 4-tuples of semantic labels?
> 7. It seems that the descriptor \theta is invariant to the labels \tilde{\mathcal{L}} - why is this the case? Why do we need the label set as an argument of f?
> - Thanks to your questions, we find some problems with the original formulation of the SIM(3)-invariant shape descriptor in the paper. So we have added more details from lines 166 to 177 in the revised paper, where the answers to the three questions will also be found. You can check the following description for detail:
> - $\theta$ denotes a function which takes 4 points as input and a SIM(3)-invariant scalar as output.
> - We call $\theta$ the atomic descriptor. However, only a single atomic descriptor calculated from 4 points (primitives centers) is not enough to describe an object's shape. Thus, we design the SIM(3)-invariant shape descriptor through a function $f$, which concatenates multiple atomic descriptors to form a SIM(3)-invariant shape descriptor. With our estimated primitives $\tilde{\mathcal{C}}$ and their semantic labels $\tilde{\mathcal{L}}$, the set $\tilde{\mathcal{L}}^4$ is denoted as:
>
>   $\tilde{\mathcal{L}}^4 = \\{(i,j,k,o)\mid i,j,k,o \in \tilde{\mathcal{L}}, i\neq j, k\neq o \\}$
>
> - and the SIM(3)-invariant shape descriptor $f$ is denoted as:
>
>   $ f(\tilde{\mathcal{C}},\tilde{\mathcal{L}}) = \underset{(i,j,k,o) \in \tilde{\mathcal{L}}^4_{sub}}
>   {\oplus}\theta(\boldsymbol{c}_i,\boldsymbol{c}_j,\boldsymbol{c}_k,\boldsymbol{c}_o)$
>
>   where $\oplus$ denotes the concatenate operation which concatenates multiple scalars into a vector in order, and $\tilde{\mathcal{L}}^4_{sub}$ is a subset randomly chosen from $\tilde{\mathcal{L}}^4$ for each optimization. We choose a subset $\tilde{\mathcal{L}}^4_{sub}$ from $\tilde{\mathcal{L}}^4$ because $|\tilde{\mathcal{L}}^4|$ would be large, which is computational expensive. An ablation is conducted to analyze the effect of $|\tilde{\mathcal{L}}^4_{sub}|$, which indicates the number of atomic descriptors in a shape descriptor.
>
> 8. Sect 3.5: "We sample 1024 points from..." - do you sample the points uniformly at random, or by another method? I suggest to try in the future farthest point sampling to guarantee uniform coverage of the input.
> - Yes, we sample 1024 points at random by numpy.random.choice, same as SGPA for equal comparison. According to your suggestion, we have tried FPS and find little influence on our method.
>
> 9. When referencing the supplementary material in the main paper, it would be helpful to always indicate a specific section in the supplement.
> - Thanks, we have fixed this issue.
> 10. \tilde{l}_i in Eq. (3) is not defined. Is this the label predicted by the network?
> - Yes, we have added this description in our revision.

---

### Official Review · Reviewer_jxdo · 2022-07-22

**Originality:** Good
**Technical Quality:** Fair
**Clarity Of Presentation:** Good
**Impact:** 3

**Recommendation:**

Weak Accept: I recommend accepting the paper, but will not argue for my recommendation if the majority of other reviewers have a different opinion.

**Summary:**

This work proposes an approach to monocular (RGB-D) category-level shape and pose estimation that decouples the two predictions through the use of a similarity-transformation-invariant descriptor. The approach uses the output of a pretrained segmentation network and operates on the resulting segmented point cloud.

The approach has two parts, first, the object shape is estimated, generatively via pretraining on ShapeNet, as both a fine-grained SDF and a course set of primitives (spheres). Then, correspondence between the generated spheres and the observed pointcloud are used to estimate scale, rotation, and translation between the predicted shape and the the robot/world/sensor.

The main novelty in this paper is the use of a descriptor that is invariant to similarity transformations as part of their shape-estimation process and the use of a course-grained shape approximation for correspondence prediction.

**Issues:**

- The authors note that too many primitives reduced performance due to 'difficulty with the classification task'. Why is this the case? Naively, it seems that because each primitive constitutes its own classification decision that more primitives would not necessarily be expected to hurt performance.

- Some of the prose in the document isn't quite grammatically correct. I recommend doing a further editing pass for such things.

**Quality Of The Limitations Section:**

Additional details required

**Reviewer Expertise:**

4: The reviewer is confident but not absolutely certain that the evaluation is correct

**Robotics Focus:**

Highly relevant to robotics but no hardware experiments

**Strengths And Weaknesses:**

Strengths:
- The SIM(3)-invariant descriptor is clever and seems well suited to this application.
- The paper provides a reasonably thorough summary of related work and the state of the field
- The authors show good performance in their experimental evaluation and well-motivate most of their design choices

Weaknesses:
- Limited ablations are performed making it difficult to determine what the takeaway from this work should be. Is the critical insight
that correspondence matching between coarse shapes proves more effective? Is the key contribution their descriptor?
- The experimental results section is quite limited
    - The qualitative results are particularly lacking and the paper would be much stronger if this were improved and more examples were added.
    - The quantitative metrics used for shape are limited (IoU). Chamfer distance seems quite appropriate here (and the authors do use it when ablating number of primitives).
    - Pose error is a bit hard to interpret as is. This would benefit significantly from also providing histograms of pose errors for each approach as well as mean and percentile metrics (e.g. 50 percentile - median error, 75th percentile error, 90th percentile error, ...)


**Summary Of Recommendation:**

I advocate for acceptance of this work due to the interesting and novel shape descriptor and I think publication would be a benefit to the community. That being said, this work would be dramatically strengthened with an improved experimental section, particularly more qualitative results.

---

> ### Author Response · Authors · 2022-08-23
> **Response to Reviewer jxdo**
>
> Thanks for your review and comments. We hope our response addresses your concerns.
>
> 1. Limited ablations are performed making it difficult to determine what the takeaway from this work should be. Is the critical insight that correspondence matching between coarse shapes proves more effective? Is the key contribution their descriptor?
> - Sorry for the confusion in the ablation section. We have reorganized our ablation in the revision. In our novel framework, semantic primitives are the basis and SIM(3)-invariant descriptors are the key insights. Therefore, the ablation verifies the effectiveness of SIM(3)-invariant descriptors for shape optimization and pose estimation, and also analyses the effect of different numbers of primitives and atomic descriptors on the results. Check ["Response to Common Concerns"](https://openreview.net/forum?id=N78I92JIqOJ&noteId=j4LblaHIZDx) for detail.
> 2. The qualitative results are particularly lacking and the paper would be much stronger if this were improved and more examples were added.
> - We have showed more qualitative results in the appendix as well as in supplementary video due to page limitations in the main paper. We would appreciate it if you could check these results in the attachment or appendix.
> 3. The quantitative metrics used for shape are limited (IoU). Chamfer distance seems quite appropriate here (and the authors do use it when ablating number of primitives).
> - Thanks for the reminder, we have added the quantitative results of the shape reconstruction in chamfer distance metric. You could check ["Response to Common Concerns"](https://openreview.net/forum?id=N78I92JIqOJ&noteId=j4LblaHIZDx) for detail, and the results are also added to our revised paper.
> 4. Pose error is a bit hard to interpret as is. This would benefit significantly from also providing histograms of pose errors for each approach as well as mean and percentile metrics (e.g. 50 percentile - median error, 75th percentile error, 90th percentile error, ...)
> - We also showed the detailed curve in the appendix and the attachment. We would also appreciate it if you could check the result.
> 5. The authors note that too many primitives reduced performance due to 'difficulty with the classification task'. Why is this the case?
> - "Classification task" refers to the part segmentation network. In fact, the part segmentation network predicts a class label for each point of the input point cloud. When the number of primitives increases, the number of labels to be predicted by the part segmentation network increases and the parameter space of the network becomes larger, which leads to poor performance of the part segmentation and affects the final pose estimation results.
> 6. Some of the prose in the document isn't quite grammatically correct. I recommend doing a further editing pass for such things.
> - Thanks for the detailed reading, we have fixed this issue in the revision.

---

> > ### Comment · Reviewer_jxdo · 2022-08-26
> > **Response to Author's comments**
> >
> > Thank you for the detailed response. While it is unfortunate that some of the information that is critical to proper context for the paper is in the appendix, I understand that space limitations in the main text are unavoidable. I would still strongly recommend moving some additional detail about pose error to the main text if at all possible.

---

> > > ### Author Response · Authors · 2022-08-27
> > > **Thanks for the advice**
> > >
> > > Thanks for the advice, and we would add more details into the main paper in the next revision.

---

### Official Review · Reviewer_4hYs · 2022-07-25

**Originality:** Very Good
**Technical Quality:** Very Good
**Clarity Of Presentation:** Very Good
**Impact:** 4

**Recommendation:**

Weak Accept: I recommend accepting the paper, but will not argue for my recommendation if the majority of other reviewers have a different opinion.

**Summary:**

This paper proposes a category-level shape and pose estimation with semantic primitives. The semantic primitives are learned to represent general shape information of objects within the same category. Then SIM(3) invariant descriptors are learned to optimize the shape of each object. The main contributions are as follow:

1. A category-level pose and shape estimation framework that uses semantic primitives to encode object shapes.
2. A SIM(3) invariant descriptors to optimize the 3D shape of the object.

The proposed method is evaluated on the NOCS dataset and shown to perform better on real data than the state-of-the-art methods.

**Issues:**

* The IOU50 in Table 2 seems to decrease when the number of primitives increases. Any idea why this is the case?
* It looks like the symmetric loss assumes the objects to have at most one symmetric axis. However, many objects have more than 1 axis of symmetry. (For example spheres.) I wonder how the proposed method deals with this issue?
* How is RANSAC implemented to reduce the shape reconstruction error as mentioned in Section 4.2?
* From my perspective, it is a little unfair to claim the learnable parameters to be only 2.3 M from the part segmentation network. Why is this the case? I would appreciate it if the authors could elaborate on this more.
* Section 4.3, Shape Optimization for Pose Estimation is a little confusing. What is the difference between mean, random, and optimized methods?
* The objects seem to have similar sizes in the dataset. Does the proposed method work well if the sizes of the objects differ drastically? (For example, a wine bottle v.s. a 12oz coke bottle.)
* This might be nit-picking but I'm curious about how the proposed method works with category ambiguity. For example, a laptop can have two forms (open and close, and potentially any pose between the two.), but they all belong to the laptop category. Will the proposed method work well in this case?


**Quality Of The Limitations Section:**

Additional details required

**Reviewer Expertise:**

3: The reviewer is fairly confident that the evaluation is correct

**Robotics Focus:**

Highly relevant to robotics but no hardware experiments

**Strengths And Weaknesses:**

**Strengths**
* The proposed method is able to reconstruct the shape and pose of the objects with only a RGB-D image.
* Semantic primitives are adopted to allow the network to learn a general representation of objects within one category.
* A SIM(3) invariant descriptor is designed to help the shape estimation process by decoupling the poes estimation task.
* The experimental results show the proposed method performs better in the real data set than the state-of-the-art method.
* In the supplementary materials, the proposed method is evaluated on unseen objects. This demonstrates the generalizability of the proposed methods.

**Weaknesses** (Also see issues)
* The symmetric loss appears to require extra annotating on the symmetric axis of each object.
* I appreciate the ablation study section. However, no experiments are designed to show the effectiveness of the Semantic primitives and the SIM(3) invariant descriptors.
* Some questions need to be clarified. (See issues)

**Summary Of Recommendation:**

This paper proposes a category-level shape and pose estimation framework. The semantic primitives and SIM(3) invariant descriptors are not new in robotics. However, it is interesting to see the combination of the two ideas for pose and shape estimation. The experimental results show the proposed method performs better on real datasets compared to the state-of-the-art methods. In addition, the generalizability of the method is demonstrated by evaluating a different dataset. However, there exist some issues that need to be addressed. Thus, I recommend a weak accept.

---

> ### Author Response · Authors · 2022-08-23
> **Response to Reviewer 4hYs (1/2)**
>
> Thanks for your review and comments. We hope our response addresses your concerns. The response contains two parts. This is the first part.
> 1. The symmetric loss appears to require extra annotating on the symmetric axis of each object. It looks like the symmetric loss assumes the objects to have at most one symmetric axis. However, many objects have more than 1 axis of symmetry. (For example spheres.) I wonder how the proposed method deals with this issue?
> -  We used CAD models from ShapeNet, most categories in which are Y-axis symmetric. And the axis required by symmetric loss is a common limitation in most category-level works.
> - For symmetric objects with one symmetric axis, we sampled $k$ equivalent cases around the symmetry axis. For $n$ symmetry axes, we could sample equivalent cases by sampling around each axis, and the number of equivalent cases becomes $k^n$. Although "sphere" has an infinite number of symmetry axes, in 3D space, the Euler angles (3 axes) can represent all rotations. Therefore $n$ should not be greater than 3 in 3D space, and as the number of symmetry axes increases, the equivalent cases become at most $k^3$, which is still computationally feasible.
> 2. No experiments are designed to show the effectiveness of the Semantic primitives and the SIM(3)-invariant descriptors.
> - Sorry for the misleading ablation. We have reorganized our ablation to justify the effectiveness of SIM(3)-invariant descriptor-based shape optimization on both shape and pose estimation. You can check ["Response to Common Concerns"](https://openreview.net/forum?id=N78I92JIqOJ&noteId=j4LblaHIZDx) for further detail. As for semantic primitives, we think it is hard to verify their effectiveness through experiments, but semantic primitives are the basis of our work.
> 3. Section 4.3, Shape Optimization for Pose Estimation is a little confusing. What is the difference between mean, random, and optimized methods?
> - Our estimated shape is recovered from the latent space. Denoting the shape embedding in the latent space as $z$,  "mean" refers to the shape when shape embedding $z=\boldsymbol{0}$; "random" refers to the shape when $z$ is a random variable $z \sim \mathcal{N}(0,1)$; "optimized" is our truly estimated shape via gradient descent in an online iterative way by minimizing the descriptor difference between the observation and estimated shape. We have added the above descriptions in the revision.
> 4. The IOU50 in Table 2 seems to decrease when the number of primitives increases. Any idea why this is the case?
> - Although fewer primitives lack geometric detail, the IoU50 is a rough metric with little influence from shape. A fewer number of primitives means better performance of the part segmentation network, so IoU50 is higher when there are fewer primitives. Moreover, it also can be seen from the ablation (as also shown in the following table) that the difference among the "mean", "random" and "optimized" canonical shapes in the IoU50 metric is slight.
>   | Shape     | IoU50 | IoU75 | $5^\circ2cm$ |$5^\circ5cm$ | $10^\circ2cm$ | $10^\circ5cm$ |
>   | --------- | ----- | ----- | ---- | ---- | ----- | ----- |
>   | Mean      | 77.7  | 54.5  | 31.7 | 36.0 | 51.4  | 61.0  |
>   | Random    | 76.8  | 51.6  | 28.7 | 34.4 | 47.6  | 60.3  |
>   | Optimized | 77.9  | 64.4  | 43.2 | 50.5 | 63.1  | 75.8  |
> 5. How is RANSAC implemented to reduce the shape reconstruction error as mentioned in Section 4.2?
> - Our shape descriptor is concatenated by multiple but not all atomic descriptors. Because the total number of atomic descriptors would be large, which is computational expensive. To reduce outliers caused by imperfect part segmentation and computational cost, we randomly selected a certain number of atomic descriptors to form the shape descriptor for each online shape optimization. This random selection is repeated several times, and the one with the highest number of atomic descriptors inliners is selected as the result. The above is what we call RANSAC.

---

> > ### Comment · Reviewer_4hYs · 2022-08-26
> > **Response to the Authors' Comments**
> >
> > I appreciate your detailed response. All of my concerns have been addressed and the explanation of the learnable parameters makes sense. I also appreciate the newly added ablation study section, as it provides insights to the proposed methods and addresses some of the concerns that I had.
> >
> > I know there're only limited pages available, so I recommend the authors to add the laptop examples in the supplementary materials.

---

> > > ### Author Response · Authors · 2022-08-27
> > > **Thanks for the advice**
> > >
> > > We are glad that we have addressed your concerns, and we would take the advice into account in the next revision.

---

> ### Author Response · Authors · 2022-08-23
> **Response to Reviewer 4hYs (2/2)**
>
> This is the second part.
>
> 6. From my perspective, it is a little unfair to claim the learnable parameters to be only 2.3 M from the part segmentation network. Why is this the case? I would appreciate it if the authors could elaborate on this more.
> - We think this is a fair and meaningful comparison, and we are glad to have a further discussion with you. First, only these 2.3M parameters were trained for pose estimation on the training dataset provided by NOCS. Except for these 2.3M parameters, the SIM(3)-invariant descriptors and pre-trained DualSDF models are not used in this training phrase. In Table. 1, we make a fair comparison with iCaps, which also uses generative models (DeepSDF) to refine the shape and pose, but they still need 80.9M parameters to compute the object poses in addition to those of DeepSDF.
> - Second, from our motivation, we believe that this comparison makes sense. Much of the existing works (e.g., large language models or generative models) provide rich prior knowledge for downstream tasks, e.g., 3D generative models provide sufficient shape priors for category-level pose estimation. One of our motivations is to take full advantage of this prior knowledge, an idea similar to the recent work "Language Models as Zero-Shot Planners" [a] which guide robots on downstream tasks by adopting large language models, like GPT-3 [b]. In our method, to perform shape and pose estimation, all we need to do is to learn a "link" between the observation and prior knowledge. The part segmentation network (2.3M) and SIM(3)-invariant shape descriptors play the role of "link". The part segmentation network extracts a suitable representation, and the SIM(3)-invariant descriptor measures the similarity between the observed and estimated implicit shapes. We believe our ideas will have a further impact as prior knowledge becomes more abundant.
> 7. The objects seem to have similar sizes in the dataset. Does the proposed method work well if the sizes of the objects differ drastically? (For example, a wine bottle v.s. a 12oz coke bottle.)
> - Yes, it works well. We designed our method with this in mind. Our part segmentation network uses 3D Graph Convolution Network (3D-GCN [c]), which is a scale-invariant network, and our shape descriptors are also scale-invariant. So our method will not be affected by the scale of the objects.
> 8. This might be nit-picking but I'm curious about how the proposed method works with category ambiguity. For example, a laptop can have two forms (open and close, and potentially any pose between the two.), but they all belong to the laptop category. Will the proposed method work well in this case?
> - Yes, it works well. In fact, in category-level pose estimation, such shape ambiguities as the opening and closing angle of the notebook are taken into account. We have selected several examples from the CAMERA dataset, and these results are obtained from the same checkpoint. Check the attachment for detail.
>
> References:
>
> [a] Huang W, Abbeel P, Pathak D, et al. Language models as zero-shot planners: Extracting actionable knowledge for embodied agents[J]. arXiv preprint arXiv:2201.07207, 2022.
>
> [b] Brown T, Mann B, Ryder N, et al. Language models are few-shot learners[J]. Advances in neural information processing systems, 2020, 33: 1877-1901.
>
> [c] Lin Z H, Huang S Y, Wang Y C F. Convolution in the cloud: Learning deformable kernels in 3d graph convolution networks for point cloud analysis[C]//Proceedings of the IEEE/CVF conference on computer vision and pattern recognition. 2020: 1800-1809.

---

### Author Response · Authors · 2022-08-23
**Summary of Revision**

We thank all reviewers and AC for the valuable comments. We hope our response addresses your concerns and we upload a draft with a few modifications, detailed as follows:
- We have added a more detailed description of our SIM(3)-invariant shape descriptor.
- We have reorganized our ablation study to justify the design choices of our method.
- We have added evaluation of shape reconstruction separately from pose estimation.
- We have added more discussion to our limitations, including:
  - As mentioned by reviewer 4hYs, our method needs extra annotating on the symmetric axis. And this limitation is shared by many other category-level works.
  - Our by-product "object mesh reconstrction" mentioned in the appendix does not have any influence or  contribution to our method. Future works could consider it and get the better pose estimation and mesh reconstruction results.

The changes are highlighted with blue color in the revised paper in attachment.

---

### Author Response · Authors · 2022-08-23
**Response to Common Concerns**

Thanks for your constructive reviews and comments. Below we address several common concerns and list the supplemented experiments.

In order to clarify common concerns about the ablation experiments and the shape reconstruction results, we have reorganized the ablation experiments and added additional results of shape reconstruction in the revised paper.  Specifically, we explore the following issues through quantitative experiments on the NOCS-REAL275 dataset (more details in section 4.3 of the revised paper):

1. Effectiveness of the SIM(3)-invariant descriptor. Does the descriptor-based shape optimization make the optimized shape more accurate and benefit the pose estimation?
- Yes, we verify the effectiveness of our SIM(3)-invariant descriptors from two aspects, both shape and pose estimation. We compared the shape and pose estimation results of the following three types of shape.
  - Mean, which refers to the mean shape when shape embedding $z=\boldsymbol{0}$
  - Random, which refers to a random shape when  $z\sim\mathcal{N}(\boldsymbol{0},\boldsymbol{1})$.
  - Optimized, which is our truly estimated shape via gradient descent in an online iterative way, by minimizing the descriptor difference between the observation and estimated shape.
- First, we verify that descriptor-based shape optimization improves the accuracy of shape reconstruction. The following table shows the shape estimation results in chamfer distance metric ($\times 10^{-3}$).
| Shape     | bottle | bowl | camera | can  | laptop | mug  | all  |
| --------- | ------ | ---- | ------ | ---- | ------ | ---- | ---- |
| Mean      | 9.55   | 4.39 | 12.4   | 3.03 | 6.59   | 2.26 | 6.05 |
| Random    | 11.6   | 5.49 | 16.1   | 3.74 | 7.12   | 3.63 | 7.56 |
| Optimized | 2.05   | 1.55 | 10.1   | 1.63 | 2.12   | 2.93 | 3.46 |
-  Second, we verify that descriptor-based shape optimization improves the pose estimation results. The following table shows the pose estimation results.
| Shape     | IoU50 | IoU75 | $5^\circ2cm$ |$5^\circ5cm$ | $10^\circ2cm$ | $10^\circ5cm$ |
| --------- | ----- | ----- | ---- | ---- | ----- | ----- |
| Mean      | 77.7  | 54.5  | 31.7 | 36.0 | 51.4  | 61.0  |
| Random    | 76.8  | 51.6  | 28.7 | 34.4 | 47.6  | 60.3  |
| Optimized | 77.9  | 64.4  | 43.2 | 50.5 | 63.1  | 75.8  |
2. Effect of different numbers of primitives. Semantic primitives are the basis of our method. How many primitives are better for pose estimation?
- We compare the pose estimation results of 64, 128, 256, and 512 primitives. We find 256 primitives perform best since this setting strikes a balance between geometry details and the part segmentation performance. The following table shows the pose estimation results with different numbers of primitives.
| Primitive Number | IoU50 | IoU75 | $5^\circ2cm$ |$5^\circ5cm$ | $10^\circ2cm$ | $10^\circ5cm$ |
| ---------------- | ----- | ----- | ---- | ---- | ----- | ----- |
| 64             | 80.2  | 60.8  | 35.1 | 45.1 | 56.2  | 74.4  |
| 128            | 78.7  | 64.2  | 38.9 | 47.7 | 59.7  | 74.4  |
| 256            | 77.9  | 64.4  | 43.2 | 50.5 | 63.1  | 75.8  |
| 512            | 76.1  | 62.5  | 41.4 | 49.0 | 59.9  | 73.7  |

3. Effect of different numbers of atomic descriptors. Our shape descriptor is concatenated by multiple atomic descriptors. How many atomic descriptors are more effective for shape optimization?
- By comparing the reconstruction accuracy, we find that the more atomic descriptors, the better. However, considering the computational cost, we choose at most $10^6$ atomic descriptors to form a shape descriptor in our main experiments. The following table shows the accuracy of shape estimation in chamfer distance metric ($\times 10^{-3}$) when different numbers of atomic descriptors are chosen.
| Number | w/o. RANSAC      | w. RANSAC        |
| ------ | ---------------- | ---------------- |
| $10^1$ | 6.79 $\\pm$ 6.5  | 4.38 $\\pm$ 5.41 |
| $10^2$ | 4.86 $\\pm$ 5.38 | 3.97 $\\pm$ 5.15 |
| $10^4$ | 3.68 $\\pm$ 4.34 | 3.59 $\\pm$ 4.54 |
| $10^6$ | 3.46 $\\pm$ 4.05 | 3.44 $\\pm$ 4.13 |

Besides, we evaluate the shape reconstruction accuracy in our ablation and make a fair comparison with SGPA in chamfer distance ($\times 10^{-3}$) on REAL275 dataset. Check the following table for details. Even without shape loss to supervise our method, our coarse shape (semantic primitives) reconstruction still shows comparable results with SGPA.
| methods   | bottle | bowl | camera | can  | laptop | mug  | all  |
| --------- | ------ | ---- | ------ | ---- | ------ | ---- | ---- |
| SGPA      | 2.93   | 0.89 | 5.51   | 1.75 | 1.62   | 1.12 | 2.44 |
| Mean      | 9.55   | 4.39 | 12.4   | 3.03 | 6.59   | 2.26 | 6.05 |
| Random    | 11.6   | 5.49 | 16.1   | 3.74 | 7.12   | 3.63 | 7.56 |
| Optimized | 2.05   | 1.55 | 10.1   | 1.63 | 2.12   | 2.93 | 3.46 |

We would be more than happy to discuss any further questions!

---

### Meta-Review · Area_Chair_3VNp · 2022-08-09

**Recommendation:** Accept (Poster)
**Confidence:** 5

**Metareview:**

The paper introduces a new method for category-level object shape and pose estimation. The novelty of the work is on introducing a semantic primitive representation with SIM(3)-invariant descriptors that can be used for object category shape and pose estimation. Experiments conducted on the NOCS dataset and the real world verify the efficiency of the proposed method.

There are several concerns about the experimental evaluation from the reviewers including more ablation studies. The shape reconstruction is usually evaluated by Chamfer distance. The method did not evaluate shape reconstruction separately from pose estimation.

The authors have successfully addressed the concerns from the reviewers during the rebuttal. The authors are encouraged to revise the final paper accordingly.

**Best Paper Nomination:**

No